# Efficient and Information-preserving future frame prediction and beyond

**Wei Yu**[1], **Yichao Lu**[1], **Steve Easterbrook**[1], **Sanja Fidler**[1,2,3]
[1]Department of Computer Science, University of Toronto
[2]Vector Institute, Canada
[3]NVIDIA
`{gnosis,yichao,sme,fidler}@cs.toronto.edu`

## Abstract

Applying resolution-preserving blocks is a common practice to maximize information preservation in video prediction, yet their high memory consumption greatly limits their application scenarios. We propose CrevNet, a Conditionally Reversible Network that uses reversible architectures to build a bijective two-way autoencoder and its complementary recurrent predictor. Our model enjoys the theoretically guaranteed property of no information loss during the feature extraction, much lower memory consumption and computational efficiency. The lightweight nature of our model enables us to incorporate 3D convolutions without concern of memory bottleneck, enhancing the model's ability to capture both short-term and long-term temporal dependencies. Our proposed approach achieves state-of-the-art results on Moving MNIST, Traffic4cast and KITTI datasets. We further demonstrate the transferability of our self-supervised learning method by exploiting its learnt features for object detection on KITTI. Our competitive results indicate the potential of using CrevNet as a generative pre-training strategy to guide downstream tasks.

## 1 Introduction

Deep learning has enjoyed tremendous success in recent years due to its ability to capture complex dependencies and non-linearities in large datasets (Krizhevsky et al. (2012); He et al. (2016); Gomez et al. (2017)). Excellent performance has been achieved on a wide range of supervised machine learning tasks, ranging from image classification (He et al. (2016)) and object detection (Ren et al. (2015)) to speech recognition (Amodei et al. (2016)). Despite the significant breakthrough in supervised learning, the potential of applying deep architectures to unsupervised learning problems remains largely unexplored. Lately there has been a surge of interest in the task of video prediction, i.e., to predict future frames of a video sequence (Wang et al. (2017; 2018); Denton et al. (2017); Denton & Fergus (2018); Villegas et al. (2017); Lee et al. (2018)). The significance of video prediction primarily lies in its potential of discovering dynamics in the physical world. The self-supervised nature of video prediction aligns well with how humans learn, without requiring large amounts of labeled data. In addition, videos can provide an abundant and virtually unlimited source of visual information. This allows video prediction models to serve as a generative pre-training strategy of feature representation learning for a variety of downstream supervised tasks.

To date, most of the existing models for video prediction employ a hybrid of convolutional and recurrent layers as the underlying architecture (Wang et al. (2017); Shi et al. (2015); Lotter et al. (2016)). Such architectural design enables the model to simultaneously exploit the ability of convolutional units to model spatial relationships and the potential of recurrent units to capture temporal dependencies. Despite their prevalence in the literature, classical video prediction architectures suffer from two major limitations. Firstly, in dense prediction tasks such as video prediction, models are required to make pixel-wise predictions, which emphasizes the demand for the preservation of information through layers. Prior works attempt to address such demand through the extensive use of resolution-preserving blocks (Wang et al. (2017; 2018); Kalchbrenner et al. (2016)). Nevertheless, these resolution-preserving blocks are not guaranteed to preserve all the relevant information, and they greatly increase the memory consumption and computational cost of the models. The second

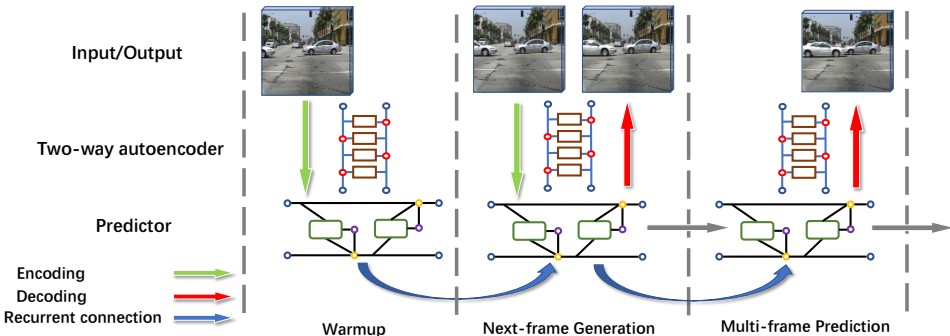

Figure 1: The pipeline of our proposed CrevNet where a single two-way autoencoder serves as both encoder and decoder. CrevNet first observes a warm-up video sequence and then starts a multi-frame video prediction without refeeding its own predictions.

drawback of existing video prediction models is that they cannot efficiently take advantage of 3D convolutions, as that would make these already cumbersome architectures even larger. 3D convolutions have been shown to be a very effective alternative to RNNs to capture temporal relations in a variety of video tasks (Liu et al. (2018); Carreira & Zisserman (2017)), and thus desirable to exploit.

Recently, reversible architectures (Dinh et al. (2014); Gomez et al. (2017); Jacobsen et al. (2018)) have attracted attention due to their light memory demand and their information preserving property by design. However, the effectiveness of reversible models remains greatly unexplored in the video literature. In this paper, we introduce a novel, conditionally reversible video prediction model, CrevNet, in the sense that when conditioned on previous hidden states, it can exactly reconstruct the input from its predictions. The contribution of this work can be summarized as follows:

- We introduce a two-way autoencoder that uses the forward and backward passes of an invertible network as encoder and decoder (Fig 1). The volume-preserving two-way autoencoder not only greatly reduces the memory demand and computational cost, but also enjoys the theoretically guaranteed property of no information loss. The lightweight nature of our model enables us to incorporate 3D convolutions without concern of memory bottleneck.

- We propose the reversible predictive module (RPM), as illustrated in Fig 2b, which extends the reversibility from spatial to temporal domain. RPM, together with the two-way autoencoder, provides a conditionally reversible architecture (CrevNet) for spatiotemporal learning. CrevNet achieves the state-of-the-art results on Moving MNIST, Traffic4cast and KITTI.

- We evaluate the effectiveness of features learnt from self-supervision by adapting our CrevNet for object detection on KITTI. Our competitive results indicate the potential of using CrevNet as a generative pre-training strategy to guide downstream CV tasks.

## 2 APPROACH

We first outline the general pipeline of our method. Our CrevNet consists of two subnetworks, an autoencoder network with an encoder $\mathcal{E}$, decoder $\mathcal{D}$ and a recurrent predictor $\mathcal{P}$ bridging encoder and decoder. Let $x_t \in \mathbb{R}^{w \times h \times c}$ represent the $t_{\text{th}}$ frame in video $x$, where $w$, $h$, and $c$ denote its width, height, and the number of channels. Given $x_{0:t-1}$, the model predicts the next frame $\hat{x}_t$ as follows:

$$\hat{x}_t = \mathcal{D}(\mathcal{P}(\mathcal{E}(x_{t-1})|x_{0:t-2})) \tag{1}$$

In the case of 3D convolution, $x_t \in \mathbb{R}^{k \times w \times h \times c}$ denotes the short video clip from $t$ to $t + k - 1$ instead of a single frame at timestep $t$, where $k$ is the temporal dimension of input or output. During the multi-frame generation process without access to the ground truth frames, the model uses its previous predictions instead.

### 2.1 THE INVERTIBLE TWO-WAY AUTOENCODER

We propose a bijective two-way autoencoder based on the additive coupling layer introduced in NICE (Dinh et al. (2014)). We begin with describing the building block of the two-way autoencoder

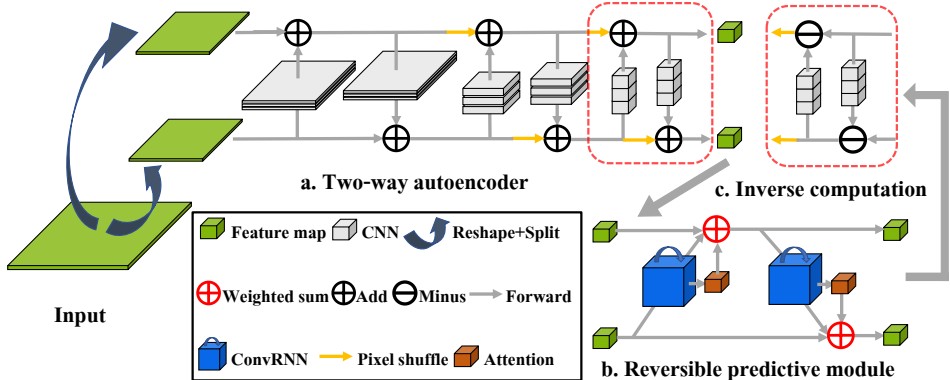

Figure 2: The network architecture of CrevNet (Better viewed in color). The input video frames are first reshaped and split channelwise into two groups. These two groups are passed to the two-way autoencoder (a) for feature extraction, and then to the predictor made up of multiple reversible predictive modules (b). The transformed high-level features produced by predictor are then passed back through the decoding pass of (a), shown here as a representative block (c) to yield its prediction.

(Fig 2a). Formally, the input $x$ is first reshaped and split channelwise into two groups, denoted as $x^1$ and $x^2$. During the forward pass of each building block, one group, e.g. $x^1$, passes through several convolutions and activations and is then added to another group, $x^2$, like a residual block:

$$\hat{x}^2 = x^2 + \mathcal{F}_1(x^1) \qquad \hat{x}^1 = x^1 + \mathcal{F}_2(\hat{x}^2) \qquad (2)$$

where $\mathcal{F}$ is a composite non-linear operator consisting of convolutions and activations, and $\hat{x}^1$ and $\hat{x}^2$ are the updated $x^1$ and $x^2$. Note that $x^1$ and $x^2$ can be simply recovered from $\hat{x}^2$ and $\hat{x}^1$ by the inverse computation (Fig 2c) as follows:

$$x^1 = \hat{x}^1 - \mathcal{F}_2(\hat{x}^2) \qquad x^2 = \hat{x}^2 - \mathcal{F}_1(x^1) \qquad (3)$$

Multiple building blocks are stacked in an alternating fashion between $x^1$ and $x^2$ to construct a two-way autoencoder, as shown in Fig 2a. A series of the forward and inverse computations builds a one-to-one and onto, i.e. bijective , mapping between the input and features. Such invertibility ensures that there is no information loss during the feature extraction, which is presumably more favorable for video prediction since the model is expected to restore the future frames with fine-grained details. To enable the invertibility of the entire autoencoder, our two-way autoencoder uses a bijective downsampling, pixel shuffle layer (Shi et al. (2016)), that changes the shape of feature from $(w, h, c)$ to $(w/n, h/n, c \times n^2)$. The resulting volume-preserving architecture can greatly reduce its memory consumption compared with the existing resolution-preserving methods.

We further argue that for generative tasks, e.g. video prediction, we can effectively utilize a single two-way autoencoder, and to use its forward and backward pass as the encoder and the decoder, respectively. The predicted frame $\hat{x}_t$ is thus given by

$$\hat{x}_t = \mathcal{E}^{-1}(\mathcal{P}(\mathcal{E}(x_{t-1})|x_{0:t-2})) \qquad (4)$$

where $\mathcal{E}^{-1}$ is the backward pass of $\mathcal{E}$. Our rationale is that, such setting would not only reduce the number of parameters in the model, but also encourage the model to explore the shared feature space between the inputs and the targets. As a result, our method does not require any form of information sharing, e.g. skip connection, between the encoder and decoder. In addition, our two-way autoencoder can enjoy a lower computational cost at the multi-frame prediction phase where the encoding pass is no longer needed and the predictor directly takes the output from previous timestep as input, as shown in Fig 1, since $\mathcal{E}(\mathcal{E}^{-1})$ is an identity mapping .

## 2.2 REVERSIBLE PREDICTIVE MODULE

In this section, we describe the second part of our video prediction model, the predictor $\mathcal{P}$, which computes dependencies along both the space and time dimensions. Although the traditional stacked-ConvRNN layers architecture is the most straightforward choice of predictor, we find that it fails to

establish a consistent temporal dependency when equipped with our two-way autoencoder through experiments. Therefore, we propose a novel reversible predictive module (RPM), which can be regarded as a recurrent extension of the two-way autoencoder. In the RPM, we substitute all standard convolutions with layers from the ConvRNN family (e.g. ConvLSTM or spatiotemporal LSTM) and introduce a soft attention (weighting gates) mechanism to form a weighted sum of the two groups instead of the direct addition. The main operations of RPM used in this paper are given as follows:

$$h_t^1 = \text{ConvRNN}(x_t^1, h_{t-1}^1) \qquad\qquad\qquad \textit{ConvRNN}$$
$$g_t = \phi(W_2 * \text{ReLU}(W_1 * h_t^1 + b_1) + b_2) \qquad \textit{Attention module}$$
$$\hat{x_t^2} = (1 - g_t) \odot x_t^2 + g_t \odot h_t^1 \qquad\qquad\quad \textit{Weighted sum}$$

where $x_t^1$ and $x_t^2$ denote two groups of features at timestep $t$, $h_t^1$ denote the hidden states of ConvRNN layer, $\phi$ is sigmoid activation, $*$ is the standard convolution operator and $\odot$ is the Hadamard product. The architecture of reversible predictive module is also shown in Fig 2b. RPM adopts a similar architectural design as the two-way autoencoder to ensure a pixel-wise alignment between the input and the output, i.e. each position of features can be traced back to certain pixel, and thus make it compatible with our two-way autoencoder. It also mitigates the vanishing gradient issues across stacked layers since the coupling layer provides a nice property w.r.t. the Jacobian (Dinh et al. (2014)). In addition, the attention mechanism in the RPM enables the model to focus on objects in motion instead of background, which further improves the video prediction quality. Similarly, multiple RPMs alternate between the two groups to form a predictor. We call this predictor conditionally reversible since, given $h_{t-1}$, we are able to reconstruct $x_{t-1}$ from $\hat{x}_t$ if there are no numerical errors:

$$x_{t-1} = \mathcal{E}^{-1}(\mathcal{P}^{-1}(\mathcal{E}(\hat{x}_t)|h_{t-1})) \tag{5}$$

where $\mathcal{P}^{-1}$ is the inverse computation of the predictor $\mathcal{P}$. We name the video prediction model using two-way autoencoder as its backbone and RPMs as its predictor CrevNet. Another key factor of RPM is the choice of ConvRNN. In this paper, we mainly employ ConvLSTM (Shi et al. (2015)) and spatiotemporal LSTM (ST-LSTM, Wang et al. (2017)) to enable a fair comparison with baselines.

## 2.3 3D CONVOLUTIONS

3D convolutions are proposed to address the shortcomings of standard 2D convolutions. The major difference between 2D-CNNs and 3D-CNNs is that at each time step 2D-CNNs take as input one video frame, while 3D-CNNs read in and output a short video clip containing $k$ continuous video frames. By applying convolutions on the temporal dimension along with the spatial dimension, models equipped with 3D convolution filters can not only extract representative spatiotemporal features, but also learn to produce consistent video clip at each generation, which further improve the quality of long-term prediction. In some cases, e.g. sequences are too short, we will use 2 consecutive frames stacked in the channel dimension instead as input at each timestep to assemble a valid warm-up sequence for ConvRNN.

## 3 EXPERIMENTS

### 3.1 LONG-TERM PREDICTION—MOVING MNIST

Moving MNIST (Srivastava et al. (2015)) is a synthetically generated dataset that contains an infinite number of sequences of length 20. Each sequence shows how 2 digits move at a constant speed and bounce inside a $64 \times 64$ frame, where each handwritten digit is randomly sampled from the MNIST dataset. By assigning different initial locations and velocities to each digit, it is possible to generate an unlimited number of sequences, thus enabling us to accurately evaluate the performance of each model without the concern of data insufficiency issues. In the default setting, models are trained to predict 10 future frames after observing 10 prior frames in the sequence. Although the dynamics of Moving MNIST seems to be simple at first glance. It is quite hard to generate consistent future frames in the task of long-term prediction as digits can bounce or occlude each other frequently.

**Datasets and Setup:** The general architecture of CrevNet used on Moving MNIST is composed of a 36-layer two-way autoencoder and 8 RPMs. All variants of CrevNet are trained by using the Adam optimizer with a starting learning rate of $5 \times 10^{-4}$ to minimize MSE. The training process is stopped

after $300,000$ iterations with the batch size of 16 and evaluated with a fixed test set containing $5,000$ sequences. To ensure that all samples in the test set are unseen by the model, digits in the training set and the testing set are separately sampled from two mutually exclusive subsets of MNIST.

| Model | MNIST-2 | | | MNIST-3 | | Memory | FLOPS |
| --- | --- | --- | --- | --- | --- | --- | --- |
| | SSIM | MSE | Human | SSIM | MSE | (*1 sample*) | (*1 frame*) |
| ConvLSTM (Shi et al. (2015)) | 0.707 | 103.3 | 0.923 | 0.695 | 127.3 | 1043 MB | 107.4 G |
| FRNN (Oliu et al. (2018)) | 0.819 | 68.4 | 0.848 | 0.791 | 90.4 | 717 MB | 80.1 G |
| VPN (Kalchbrenner et al. (2016)) | 0.870 | 70.0 | 0.831 | 0.820 | 85.6 | 5206 MB | 309.6 G |
| PredRNN (Wang et al. (2017)) | 0.869 | 56.8 | 0.837 | 0.822 | 83.1 | 1666 MB | 192.9 G |
| PredRNN++ (Wang et al. (2018)) | 0.898 | 46.5 | 0.781 | 0.864 | 68.4 | 2017 MB | 106.8 G |
| E3D-LSTM (Wang et al. (2019)) | 0.910 | 41.3 | 0.706 | 0.870 | 62.4 | 2695 MB | 381.3 G |
| CrevNet + ConvLSTM | 0.928 | 38.5 | 0.602 | 0.886 | 57.2 | **130 MB** | 0.919 G |
| CrevNet + ST-LSTM | **0.949** | **22.3** | **0.558** | **0.916** | **40.6** | 195 MB | 1.618 G |

Table 1: Quantitative evaluation of different methods on Moving MNIST. All metrics are averaged over the 10 predictions. Lower MSE and higher SSIM indicates better prediction accuracy.

We compare CrevNet to six popular benchmark models from the literature: (i) ConvLSTM (Shi et al. (2015)), (ii) FRNN (Oliu et al. (2018)), (iii) VPN (Kalchbrenner et al. (2016)), (iv) PredRNN (Wang et al. (2017)) , (v) PredRNN++ (Wang et al. (2018)), and (vi) E3D-LSTM (Wang et al. (2019)),. All baselines are implemented and optimized by following their corresponding protocols. To test our model in a more challenging setting, we also extend Moving MNIST to a 3-digit version where digits are more likely to occlude each other.

**Results:** The performance of each model in terms of per-frame MSE and the Structural Similarity Index Measure (SSIM) (Wang et al. (2004)) is presented in Table 1. CrevNet outperforms all previous methods by a wide margin on both metrics while memory consumption of all CrevNet variants is significantly lower than that of other baselines. In particular, CrevNet with ConvLSTM only uses 130 MB memory per sample and is still capable of yielding results better than any baselines.

To analyze the contribution of each module, we conduct an ablation study on both ConvLSTM and ST-LSTM with respect to 3D convolution, two-way autoencoder and RPM and summarize the results in Table 2. Note that we do not include the quantitative results of the combination of two-way autoencoder and stack-ConvRNN predictor because it fails to produce consistent long-term generations and we choose UNet (Ronneberger et al. (2015)) as an alternative to our two-way autoencoder. We can observe a significant improvement over ConvLSTM after we embed it into our CrevNet framework, indicating the effectiveness of reversible architectures. Also, integrating 3D convolution can consistently enhance the performance of all architectures. To further show the superior performance of CrevNet, we evaluate it on a harder 3-digit setting. Results are shown in the right column of Table 1. Compared with the 2-digit setting, all models suffer a deterioration in quantitative performance due to the more frequent occurrence of overlapping digits. Nevertheless, our CrevNet still achieves the best result.

Figure 3: An extremely hard sequence of Moving MNIST where two digits are continuously overlapped during the warm-up phase.

| Model | ConvLSTM | | ST-LSTM | |
| --- | --- | --- | --- | --- |
| | MSE | SSIM | MSE | SSIM |
| Stacked-RNN | 103.3 | 0.707 | 56.8 | 0.869 |
| 3D Stacked-RNN | 85.8 | 0.785 | 46.2 | 0.878 |
| UNet+StackedRNN | 83.5 | 0.793 | 58.8 | 0.865 |
| UNet+RPM | 63.4 | 0.855 | 50.2 | 0.896 |
| CrevNet w/o 3D | 50.2 | 0.888 | 40.4 | 0.916 |
| CrevNet + 3D | 38.5 | 0.928 | 22.3 | 0.949 |

Table 2: An ablation study w.r.t. 3D convolution, RPM and two-way autoencoder. All metrics are averaged over the 10 predictions.

In Fig 3 , our qualitative analysis shows how each model performs on an extremely hard case of Moving MNIST where two digits are continuously overlapped during the warm-up phase.

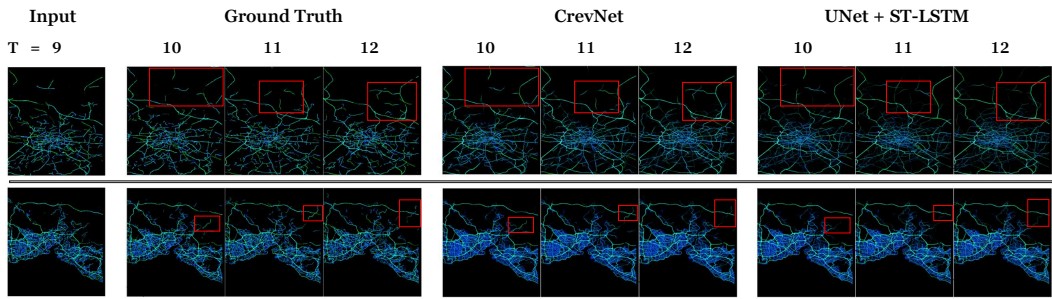

Figure 4: The visual comparison of Traffic4cast. The red boxes track some dynamics successfully captured by our CrevNet. Better viewed large.

As we can see, our model is the only model that can differentiate the overlapping digits. The information-preserving property of the two-way autoencoder enables our method to reconstruct every fine detail of moving digits after occlusion while baselines typically only restore the basic shape of these numbers. In fact, our CrevNet works almost perfectly on Moving MNIST, with most of its generations being visually indistinguishable from groundtruth.

We perform a human study to assess the fidelity of the video clips generated by different models. We presented pairs of video clips to human judges, where each pair consists of a video clip from the test set together with the prediction generated by the model. The judges were asked to decide which of the two video clips is more likely to be the groundtruth. To make each trail blind, the judges were not informed which model is used for generation and two sequences were randomly displayed on either side of the screen. We totally collected 2439 responses made by 58 human subjects and then calculated the probability that human judges answered correctly. The results are reported in Table 1. The accuracy of 55.8 % suggests that subjects could hardly detect the difference and their decisions were very close to random guesses.

## 3.2 SHORT-TERM PREDICTION—TRAFFIC FLOW FORECASTING

Next, we evaluate our model on a more complicated real-world dataset, Traffic4cast (IARAI (2019)), which collects the traffic statuses of 3 big cities over a year at a 5-minute interval. Traffic forecasting can be straightforwardly defined as video prediction task by its spatiotemporal nature. However, this dataset is quite challenging for the following reasons. (1). High resolution: The frame resolution of Traffic4cast is $495 \times 436$, which is the highest among all datasets. Existing resolution-preserving methods can hardly be adapted to this dataset since they all require extremely large memory and computation. Even if these models can be fitted in GPUs, they still do not have large enough receptive fields to capture the meaningful dynamics as vehicles can move up to 100 pixels between consecutive frames. (2). Complicated nonlinear dynamics: Valid data points only reside on the hidden roadmap of each city, which is not explicitly provided in this dataset. Moving vehicles on these curved roads along with tangled road conditions will produce very complex nonlinear behaviours. It also involves many unobservable conditions or random events like weather and car accidents.

**Datasets and Setup:** Each frame in Traffic4cast dataset is a $495 \times 436 \times 3$ heatmap, where the last dimension records 3 traffic statuses representing volume, mean speed and major direction at given location. The architecture of CrevNet is the same as the one we used on Moving MNIST. As we mentioned before, the existing resolution-preserving methods cannot handle such high resolution input. Thus, to make the comparison possible, we add U-Net encoder-decoder to the baseline models including ConvLSTM and ST-LSTM. We train each model to predict next 3 frames (the next 15 minutes) from 9 observations and evaluate prediction with MSE criterion.

| Model | CrevNet | UNet+ST-LSTM | UNet+ConvLSTM | Best Team | 2nd Best Team |
|-------|---------|--------------|---------------|-----------|---------------|
| MSE | $9.340 \times 10^{-3}$ | $9.725 \times 10^{-3}$ | $9.846 \times 10^{-3}$ | $9.559 \times 10^{-3}$ | $9.717 \times 10^{-3}$ |

Table 3: Quantitative evaluation on Traffic4cast. Lower MSE indicates better prediction accuracy.

Input                                    Predictions (T = 11)

T = 10          Groundtruth        CrevNet        ContextVP        PredNet

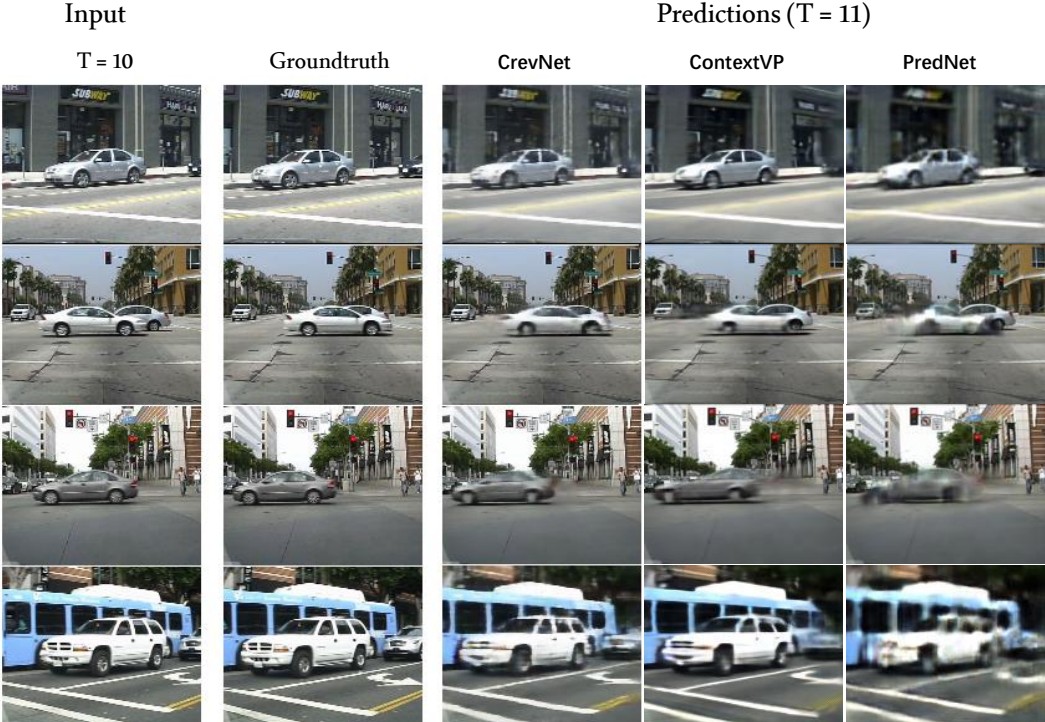

Figure 5: The visual comparison of next-frame predictions on Caltech Pedestrian.

**Results:** The quantitative comparison including the best two results on the leaderboard before the submission of this paper is provided in Table 3. Unlike all previous state-of-the-art methods, CrevNet does not suffer from high memory consumption so that we were able to train our model in a single V100 GPU. The invertibility of two-way autoencoder preserves all necessary information for spatio-temporal learning and allows our model to generate sharp and reasonable predictions. As illustrated in Fig 4, our model can identify and remember the hidden roadmap of each city through the learning of complicated nonlinear dynamics and accurately predict how traffic system will evolve.

## 3.3    NEXT-FRAME PREDICTION AND BEYOND—CAR-MOUNTED CAMERA VIDEO

The real-world videos are usually long-term unpredictable because of the intrinsic randomness and the lack of necessary information. Thus, the common practice for datasets like KITTI (Geiger et al. (2012)), a car-mounted camera video dataset, is to perform next-frame prediction. In this section, we further demonstrate the superior performance of our CrevNet by conducting experiments on KITTI and Caltech Pedestrian (Dollár et al. (2009)). Compared with the previous two settings, car-mounted camera videos dataset presents another level of difficulty for video prediction as it describes various nonlinear three-dimensional dynamics of multiple moving objects including backgrounds. Furthermore, as our well-trained model is capable of generating authentic future frames, it should spontaneously learn at least the shape and location of all moving objects, which indicates that the learnt features are very informative for downstream tasks. For example, in the case of object detection, these features can be incorporated to estimate more accurate locations and sizes of bounding boxes. Therefore, we also explore the effectiveness of our self-supervised learning method on the 2D object detection on KITTI.

### 3.3.1    VIDEO PREDICTION

**Datasets and Setup:** We follow the same protocol used in PredNet (Lotter et al. (2016)) for preprocessing and evaluation. We first center-crop all video frames and resize them into $128 \times 160$. We compare our proposed method with 4 state-of-the-art benchmark models. Models are trained on KITTI dataset to predict the next frame after 10-frame warm-up and are evaluated on Caltech Pedestrian. The architecture of CrevNet used on KITTI is composed of a 48-layer two-way autoencoder

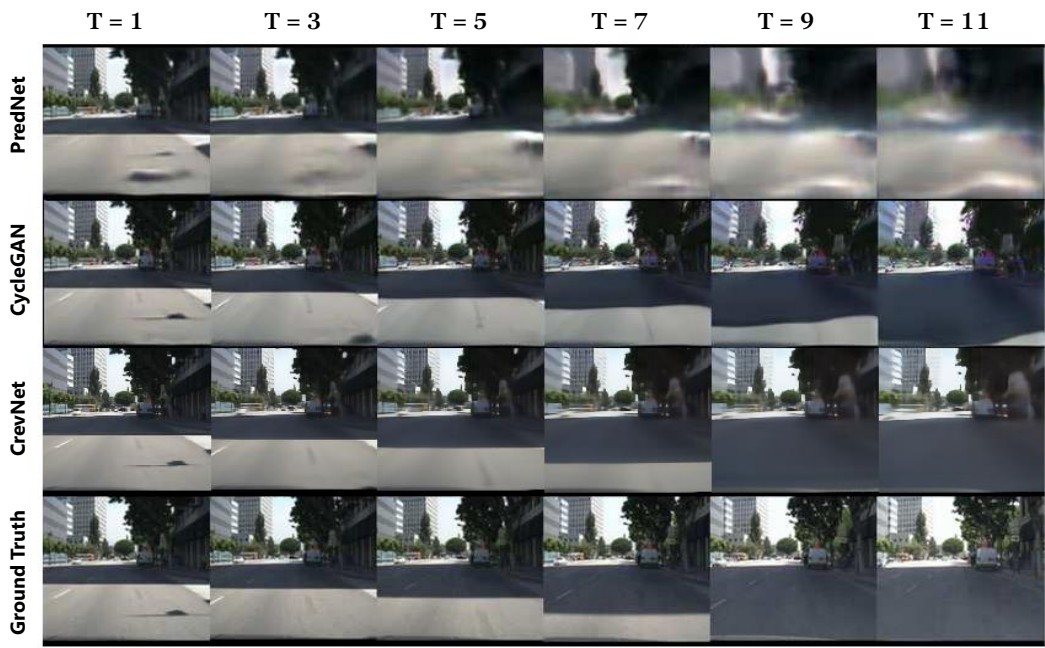

Figure 6: The visual comparison of 12-frame prediction on Caltech Pedestrian. Notice how well CrevNet captures the detail and geometry of the buildings in the background, and the overall shading.

and 40 RPMs. Note that it is the memory efficiency of our method that allows us to deploy such deep model. Since our model also possess good capability of long-term prediction. We add a 12-frame prediction comparison with CycleGAN (Kwon & Park (2019)) and PredNet (Lotter et al. (2016)).

| Model | Next-Frame | | 3rd | 6th | 9th | 12th | Average |
|---|---|---|---|---|---|---|---|
| | PSNR | SSIM | | | SSIM | | |
| Copy-Last-Frame | 23.3 | 0.779 | —— | —— | —— | —— | —— |
| Dual Motion GAN (Liang et al. (2017)) | —— | 0.899 | —— | —— | —— | —— | —— |
| ContextVP (Byeon et al. (2018)) | 28.7 | 0.921 | —— | —— | —— | —— | —— |
| PredNet (Lotter et al. (2016)) | 27.6 | 0.905 | 0.72 | 0.66 | 0.61 | 0.58 | 0.701 |
| CycleGAN (Kwon & Park (2019)) | 29.2 | 0.919 | 0.83 | 0.73 | 0.67 | 0.63 | 0.752 |
| CrevNet | **29.3** | **0.925** | **0.84** | **0.76** | **0.70** | **0.65** | **0.776** |

Table 4: Quantitative evaluation of different methods on the Caltech Pedestrian dataset. Higher PSNR or SSIM means better prediction accuracy.

**Results:** Performance of different models in terms of PSNR and SSIM is displayed in Table 4. CrevNet outperforms all baselines in both next-frame and multi-frame prediction regimes. Visual comparisons are provided in Fig 5 and Fig 6. Especially, in the case of 12-frame generation, we can observe that compared with our method, PredNet suffers severely from the famous error propagation of RNN issue while CycleGAN produces realistic yet physically inconsistent predictions.

### 3.3.2 2D OBJECT DETECTION

**Datasets and Setup:** KITTI provides three prior frames of unlabeled data for each labeled image. This allows us to run our CrevNet to extract useful spatiotemporal features for object detection. All video sequences were recorded at 10 Hz with resolution of $1242 \times 375$. We first resize each frame to $416 \times 128$ and finetune our best model on the video prediction task solely. The combinations of features extracted by our two-way autoencoder and attention masks of the target frame are then fed into the detection head for the further training. Note that we do not update the weights of CrevNet at this stage to purely demonstrate the power of self-supervised learning. Two image-based detection

models, SqueezeDet and RRC, are compared as baselines. We also add an experiment on transfer learning of features learnt by PredNet on KITTI as comparison. To be consistent with related work, we use SSD (Liu et al. (2016)) as detection head.

| Backbone | Add-ons | Car | | | Pedestrain | | | Cyclist | | | mAP |
| --- | --- | --- | --- | --- | --- | --- | --- | --- | --- | --- | --- |
| | | E | M | H | E | M | H | E | M | H | |
| SqueezeDet | (Wu et al. (2017)) | 90.4 | 87.1 | 78.9 | 81.4 | 71.3 | 68.5 | **87.6** | 80.3 | **78.1** | 80.4 |
| RRC | (Ren et al. (2017)) | 90.61 | 90.23 | **87.44** | 84.16 | 75.33 | 70.39 | 84.96 | 76.49 | 65.46 | 80.56 |
| Best Results on the Leaderboard | | **91.96** | **91.97** | 84.57 | 88.27 | 81.73 | 75.29 | 84.28 | 79.24 | 71.22 | 83.17 |
| PredNet | (Lotter et al. (2016)) | 59.05 | 41.61 | 37.53 | 50.88 | 47.51 | 43.44 | 46.25 | 43.79 | 38.66 | 45.41 |
| CrevNet | attention mask | 91.53 | 90.95 | 85.71 | 89.31 | 82.55 | 75.21 | 85.51 | 80.41 | 71.52 | 83.63 |
| | + extracted features | 91.94 | 91.84 | 85.97 | **89.66** | **83.17** | **75.80** | 87.33 | **80.91** | 72.21 | **84.31** |

Table 5: Quantitative evaluation of different methods on 2D KITTI detection in term of Average Precision (AP). The numbers in the "Best Results on the Leaderboard " are achieved by different best models on each class according to the default ranking mechanism on the Leaderboard.

**Results:** The results of all experiments and baselines can be found in Table 5. Surprisingly, our CrevNet even outperforms the combination of the best model on each class in term of mAP. Since our model is capable of capturing the motion information, it is sensitive to the small (hard) moving objects. However, the motion information alone is not sufficient for object detection due to the appearance of relatively static objects. Therefore, we can observe a performance boost after we incorporate the features extracted by our two-way autoencoder. Another advantage of our method is that it can provide a better localization of bounding box since the learnt features of CrevNet remain the pixel-wise alignment with the input and output frame. Finally, thanks to the lightweight nature of our CrevNet, our best detection model can run at 6.8 FPS at the testing time.

## 4 RELATED WORK

**Deep Learning in Video Prediction:** Mainstream video prediction models can mostly be categorized into two frameworks, stacked ConvRNNs and encoder-predictor-decoder models. The former framework attempts to design a new spatiotemporal module and then stacks multiple such modules to form the final model, while the latter usually utilizes an autoencoder to project video frames into their latent representations and then employs a recurrent neural network to model the temporal transformations. PredNet (Lotter et al. (2016)) is a good representative of stacked ConvRNNs framework. In PredNet, each ConvLSTM layer produces a layer-specific prediction at every time step to transmit an error term to the next layer. This model works well for predicting the next frame, but fails to maintain its performance in a long-term setting. To tackle long-term predictions, PredRNN (Wang et al. (2017)) proposed a new spatiotemporal LSTM, which allows memory to flow both vertically and horizontally. PredRNN++ (Wang et al. (2018)) further improved the results by rearranging spatial and temporal memory in a cascaded mechanism, and by using a gradient highway architecture to ease the optimization. E3D-LSTM (Wang et al. (2019)) effectively recalled the previous memory states and also proposed to include 3D convolutions to enhance its performance. ContextVP (Byeon et al. (2018)) introduced a fully context-aware architecture consisting of parallel multi-dimensional LSTM units and blending units. Methods from stacked ConvRNNs family usually yield more accurate deterministic predictions but they consume considerable GPU memory and computational power as they abandon downsampling to prevent information loss.

The encoder-predictor-decoder framework, on the other hand, provides more flexibility than its counterpart. MCNET (Villegas et al. (2017)) and DrNet (Denton et al. (2017)) decompose the content and motion in videos by building their corresponding encoders and then integrate this disentangled information to yield the next frame. Retrospective CycleGAN (Kwon & Park (2019)) combines sequential adversarial loss with frame adversarial loss, which encourages the model to generate frames that are visually similar to authentic images. In terms of modeling stochasticity, SVG (Denton & Fergus (2018)) and SAVG (Lee et al. (2018)) utilize a prior inference network to mimic the uncertainty in the environment, and then embed it into a deterministic generative model to produce stochastic video frames. VPN (Kalchbrenner et al. (2016)) estimates the discrete joint distribution of the raw pixel values in a video using the well-established PixelCNNs. It is worth noticing that VPN

employs a resolution-preserving encoder to circumvent the information loss, showing the need for an efficient information-preserving encoder in the community.

**Comparison with Related Works:** To the best of our knowledge, our CrevNet is the first conditionally reversible model in the video prediction literature. There are three prior arts, E3D-LSTM (Wang et al. (2019)), FRNN (Oliu et al. (2018)) and VideoFlow (Kumar et al. (2019)), having some similarities with our CrevNet. While E3D-LSTM also employs 3D convolutions, their implementation is essentially equivalent to applying two 2D convolutional operations, as there is no shared filter on the temporal dimension. Similar to CrevNet, FRNN reduces its computational cost by eliminating the need to re-encode the output of decoder. However, FRNN has a substantially different architecture compared to CrevNet. While the encoder and decoder in our model do not need information sharing at all, FRNN relies heavily on the sharing of the hidden states between them. Although VideoFlow also utilizes invertible transformation. This approach is very different from ours because: (1). VideoFlow is built upon Glow, a very memory-consuming architecture. Such memory limits preclude the use of 3D convolutions, or even from training the model with Adam. (2). They use ANN to model temporal relationship. As such, VideoFlow cannot capture complex dynamics. (3). So far, VideoFlow has only been applied to stochastic video generation instead of deterministic video prediction.

**The Reversible and Invertible Architectures:** The idea of the coupling layer was initially introduced in NICE (Dinh et al. (2014)) so as to make the computation of the determinant of the Jacobian and inverse Jacobian trival. Inspired by additive coupling layer, RevNet (Gomez et al. (2017)) introduced a reversible block that allowed the reconstruction of activations of each layer from that of the next layer, thus eliminating the need to store activations between downsampling and significantly reducing its memory consumption. The follow-up work by (Jacobsen et al. (2018)) further proposed an invertible extension, i-RevNet, which enabled the model to preserve all information of input through layers while still being capable of extracting a useful representation for classification.

## 5 CONCLUSION

We described a novel conditionally reversible network, CrevNet, for pixel-level prediction of future frames in videos. The originality of our model lies in our use of the reversible two-way autoencoder and the accompanying reversible predictive module. Such architectural design enables the model to preserve fine-grained information without significant memory and computation overhead. CrevNet achieves state-of-the-art results on both synthetic and real-world datasets. The subsequent detection experiments demonstrate the potential of CrevNet to be a continuous self-supervised learning system to enhance downstream CV tasks, as shown in the case of BERT (Devlin et al. (2018)) for NLP tasks.

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

# A CONVLSTM AND ST-LSTM

The key equations of ConvLSTM are shown as belows.

$$i_t = \sigma(W_{xi} * \mathcal{X}_t + W_{hi} * \mathcal{H}^l_{t-1} + b_i)$$
$$f_t = \sigma(W_{xf} * \mathcal{X}_t + W_{hf} * \mathcal{H}^l_{t-1} + b_f)$$
$$\mathcal{C}^l_t = f_t \circ \mathcal{C}^l_{t-1} + i_t \circ \tanh(W_{xc} * \mathcal{X}_t + W_{hc} * \mathcal{H}^l_{t-1} + b_c)$$
$$o_t = \sigma(W_{xo} * \mathcal{X}_t + W_{ho} * \mathcal{H}^l_{t-1} + W_{co} \circ \mathcal{C}^l_t + b_o)$$
$$\mathcal{H}^l_t = o_t \circ \tanh(\mathcal{C}^l_t)$$

where $*$ denotes the convolution operator and $\circ$ denotes the Hadamard product. Based on ConvLSTM, spatiotemporal LSTM (ST-LSTM) in PredRNN adds another vertical memory flow to enhance the long-term temporal dependency as follow.

$$i_t = \sigma(W_{xi} * \mathcal{X}_t + W_{hi} * \mathcal{H}^l_{t-1} + b_i)$$
$$f_t = \sigma(W_{xf} * \mathcal{X}_t + W_{hf} * \mathcal{H}^l_{t-1} + b_f)$$
$$\mathcal{C}^l_t = f_t \circ \mathcal{C}^l_{t-1} + i_t \circ \tanh(W_{xc} * \mathcal{X}_t + W_{hc} * \mathcal{H}^l_{t-1} + b_c)$$
$$i'_t = \sigma(W'_{xi} * \mathcal{X}_t + W_{mi} * \mathcal{M}^{l-1}_t + b'_i)$$
$$f'_t = \sigma(W'_{xf} * \mathcal{X}_t + W_{mf} * \mathcal{M}^{l-1}_t + b'_f)$$
$$\mathcal{M}^l_t = f'_t \circ \mathcal{M}^{l-1}_t + i'_t \circ \tanh(W_{xm} * \mathcal{X}_t + W_{mm} * \mathcal{M}^{l-1}_t + b_m)$$
$$o_t = \sigma(W_{xo} * \mathcal{X}_t + W_{ho} * \mathcal{H}^l_{t-1} + W_{co} \circ \mathcal{C}^l_t + W_{mo} \circ \mathcal{M}^l_t + b_o)$$
$$\mathcal{H}_t = o_t \circ \tanh(W_{1 \times 1}[\mathcal{C}^l_t, \mathcal{M}^l_t])$$

where blue part overlaps ConvLSTM. Note that $\mathcal{M}^l_t$ usually receives information from the previous layer instead of the previous state and the special case is that $\mathcal{M}^1_t$ receives $\mathcal{M}^L_{t-1}$ to constitute a zigzag information flow. As we can see, ST-LSTM basically doubles the size of feature map and the number of parameters compared with ConvLSTM.

# B CONDITIONAL REVERSIBILITY

As we mentioned in Section 2.2, conditional reversibility is an interesting property of our CrevNet. In this section, we will provide more details about it. Given $\hat{x}^2_t, x^1_t$ and $h^1_{t-1}$, the reversible predictive module can recover $x^2_t$ as follow

$$h^1_t = \text{ConvRNN}(x^1_t, h^1_{t-1})$$
$$g_t = \phi(W * h^1_t + b)$$
$$x^2_t = (\hat{x}^2_t - g_t \odot h^1_t) \odot 1/1 - g_t$$

Ideally, if there is no numerical error during the calculation, we can get the perfect reconstruction of input by applying this inverse operation repeatedly. In practice, while the most of reverse generations are successful, the inevitable numerical error will still result in some failing cases, especially in the case of a very deep architecture because errors will be amplified layer by layer.

# C OBJECT DETECTION

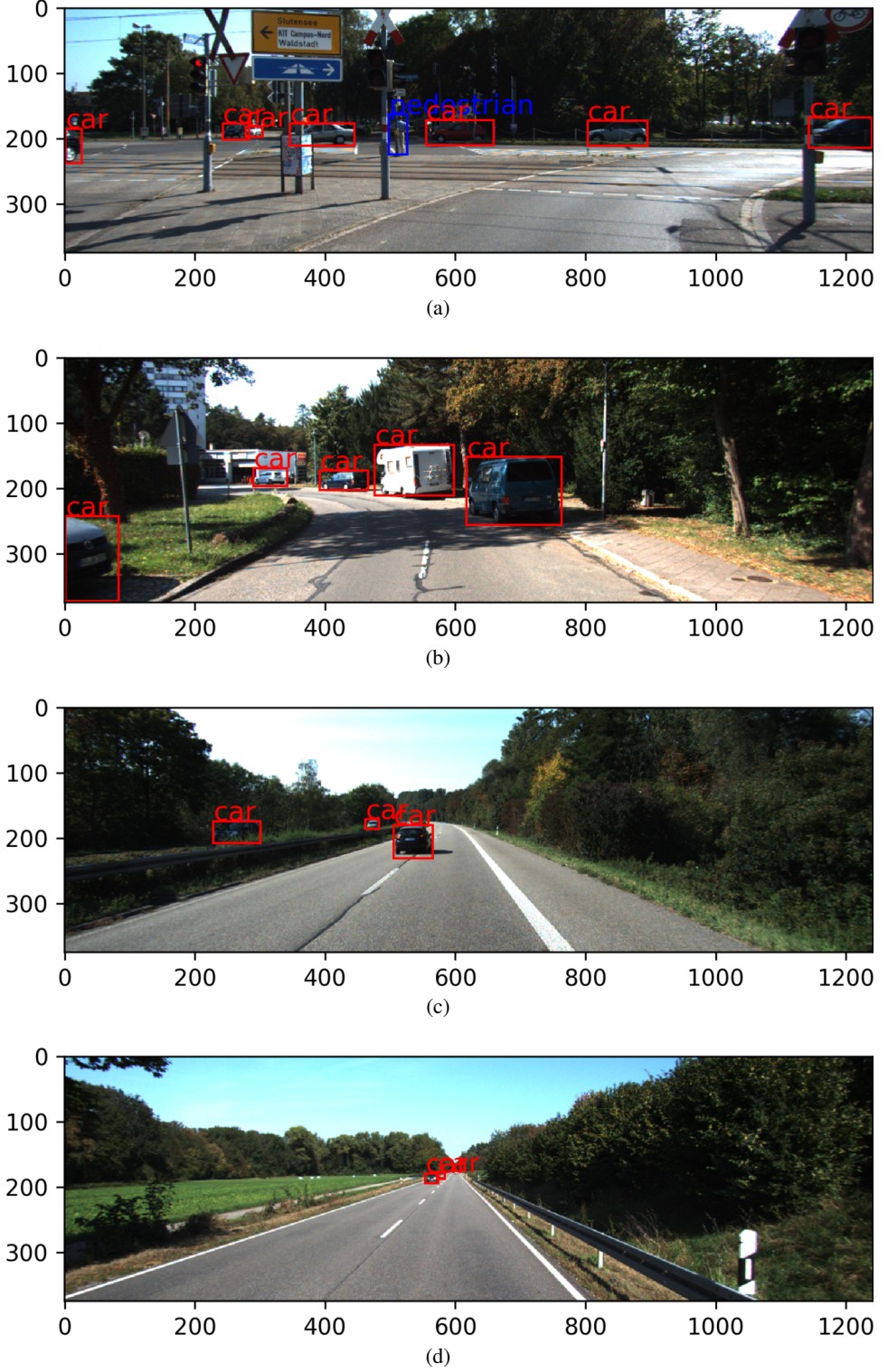

Figure 7: Visualization of 2D Object Detection on KITTI.

