# OpenReview forum: "Efficient and Information-Preserving Future Frame Prediction and Beyond"
_ICLR.cc/2020/Conference — Accept (Poster)_

### Official Review · AnonReviewer2 · 2019-10-23
**Official Blind Review #2**

**Rating:** 6

**Review:**

This paper introduces Conditionally Reversible Network (CrevNet) that consists of the invertible autoencoder and a reversible predictive module (RPM). The two-way autoencoder is an invertible network that preserves the volume with no information loss while reducing memory consumption by using bijective downsampling. The RPM is a recurrent extension of two-way autoencoder that provides the reversiblity in temporal domain. The experiments on Moving MNIST, Traffic4cast, KITTI, and 2D object detection on KITTI show the improvement compare to other state-of-the-art models.

The paper is well-written and the contributions are clear. The experiments on diverse tasks and datasets are provided. Especially, the use of pretrained generative model for the object detection task shows the benefit of the model.

However, it is not clear the necessity of the proposed model and how the memory demand and computation cost are reduced.

To have a better support for the contributions, I have following suggestions.

- The benefit or necessity of each module is not clear. The comparisons with 1) only the invertible two-way autoencoder (without Section 2.2 and 2.3)", 2) only RPM (without Section 2.1 and/or 2.3), and 3)two-way autoencoder + reversible predictive model without 3D convolutions (without Section 2.3) are required.

- Model size/memory comparison is provided only for the MNIST dataset. Since the main benefit of the proposed model is low memory demand and computation cost, I suggest to provide the the comparisons of the memory demand with other models as well.
1) Other models on on Caltech Pedestrian dataset, Especially, the performance gap on Caltech Pedestrian dataset between the proposed method and CycleGAN is very small.
2) Some of the compared models on 2D KITTI detection are also lightweighted e.g., SqeeqeDet. The comparison of the model size/memery/test time with these models are necessary.

----
<After rebuttal>
Authors have mostly addressed my concerns, and the contributions are clearer. I adjusted my rating accordingly.

**Experience Assessment:**

I have published in this field for several years.

**Review Assessment: Checking Correctness Of Derivations And Theory:**

I carefully checked the derivations and theory.

**Review Assessment: Checking Correctness Of Experiments:**

I carefully checked the experiments.

**Review Assessment: Thoroughness In Paper Reading:**

I read the paper thoroughly.

---

> ### Author Response · Authors · 2019-11-09
> **Response to Reviewer #2**
>
> Thank the reviewer for the feedback.
>
> --How do we reduce memory and computation: As we mentioned in the paper, to maximize the information preservation, most of the state-of-the-art methods (e.g. ConvLSTM, PredRNN, ContextVP, E3D-LSTM, VPN and etc) adopt the resolution-preserving design where the dimension of intermediate feature maps is w*h*k. Here, w and h will always remain the same as those of the input and k can range from 64 to 256. Our method, on the other hand, is volume-preserving, which means no matter how we change the feature maps, it will stay the same size of input, w*h*c, where c is usually 3. For example, if the input size is 128*128*3= 49152, the feature size may be 16*16*192=49152. The much smaller size of intermediate feature maps results in less memory and computation cost.
>
> --Memory demand comparison: We feel this comment may arise from a misunderstanding of the major benefit of our proposed method. The major benefit of our CrevNet is its state-of-the-art predictive power. Although memory and computational efficiency is a useful merit of our method, the key point is the information preservation property of reversible architecture that brings predictive improvement. Otherwise, we can simply adopt UNet to get memory and computation reduced as UNet is not resolution-preserving. Such efficiency also allows us to integrate other modules like 3D convolution or to apply a deeper (wider) model to get further improvement. Thus in the setting of Moving MNIST, Traffic4cast and Caltech pedestrian, our experiments are designed to show that our CrevNet can achieve the best video prediction results. We have shown that our method significantly outperformed the UNet counterpart and all baseline models.
> ------Moving MNIST and Traffic4cast: We add the computational cost comparison for Moving MNIST in Table 1. We didn’t include memory comparison on Traffic4cast because all existing state-of-the-art methods are unable to be fitted in GPUs as the simplest ConvLSTM will require ~30GB GPU memory (by our estimation) on Traffic4cast in the case of batch size of 1. As such, we cannot even fit a single ConvLSTM layer with channel size of 128 into a single GPU. We think this fact alone is enough to show the memory efficiency of our method and why our method is necessary.
> ------KITTI and Caltech pedestrian: We mainly focused on demonstrating the effectiveness of transfer learning enabled by our video prediction model in the setting of KITTI and Caltech pedestrian. We did not intend to show any superiority on speed or memory efficiency since we used 40 layers of RPM in CrevNet to achieve the best performance on video prediction. (efficiency/accuracy tradeoff) Note that it is the memory efficiency of our method that enabled us to deploy such deep model. As a comparison, the previous state-of-the-art method, Contextvp, can only be implemented in a 4-layer architecture, yet still consumes more memory and computation than our method.
> ------KITTI detection: We believe it is obvious that video-based detectors will consume more resources than light-weight image-based detectors. Nevertheless, 6.8 FPS is still faster than the best two results on the leaderboard and our transfer learning detection method even outperforms the combination of the best model for each class in terms of mAP.
> Link to leaderboard: http://www.cvlibs.net/datasets/kitti/backups/2019_10_08_10_46_41_object.html
>
> --Video prediction on Caltech pedestrian: We would argue that our improvement is non-trivial. PredNet was published 3 years ago in ICLR 2017 and subsequent improvement has been slow. After a 3-year-long effort from the community, SSIM on the Caltech pedestrian datasets was improved from 0.905 to 0.921. As you can see in Table 4, direct copying from the last frame will give us decent number (SSIM 0.779). Thus, we think an improvement of 0.004 on Caltech pedestrian  in terms of SSIM is impressive for this task, and it is reasonable for us to claim state-of-the-art. Furthermore, results on long-term prediction shows that our method performed much better than baselines in the long-term horizon. We have added the average SSIM over all 12-frame predictions in Table 4 and our method achieved an increase of 0.024 (from 0.752 to 0.776) over cycleGAN in terms of SSIM.
>
> --Benefit of each module: The ablation study of each module was provided in Table 2 and the last two rows demonstrate the effect of 3D convolution. However, the other 2 proposed comparisons will not be meaningful because:
> Only two-way autoencoder: Bare two-way autoencoder doesn’t encode any temporal information and thus cannot learn future frame prediction.
> Only RPM: This architecture doesn’t have enough model capacity (The number of channels is too small) and receptive field to learn spatiotemporal relationship. We did include UNET+RPM to demonstrate the benefit of RPM in Table 2.

---

### Official Review · AnonReviewer3 · 2019-10-23
**Official Blind Review #3**

**Rating:** 6

**Review:**

In this paper, the authors propose a new method of self-supervised feature learning from videos based on learning future frame prediction. The idea is similar as BERT like NLP tasks, but for videos, the computational cost and memory cost could be very large. To solve this problem efficiently, the authors adopt several existing techniques such as pixel shuffle layer, 3D-CNN,  ConvRNN and Attention module to efficiently and effectively capture video information. Experiments on several datasets are conducted to show the effectiveness of the proposed method.

The idea of self-supervised feature learning from videos are not novel. The key is how to learn good features that can generalize very well. The authors show that the learned features in this paper can be used on other tasks, such as object detection. And state-of-the-art results on KITTI dataset could be achieved based on the learned features with fixed backbone parameters. Although the provided results may not be state-of-the-art (for car, it seems that KITTI best results are 97%+ for easy, 95%+ for medium, 90%+ for hard, instead of 92%, 92%, and 85% provided in the paper), the generalization ability on object detection looks very interesting.

Is the generalization ability of the proposed method better that existing methods such as CycleGAN, PredNet and ContextVP? More experiments about the quality of the learned features of the proposed method are recommended to improve the importance of this paper. The advantage of the proposed method looks weak without these comparisons.

**Experience Assessment:**

I have read many papers in this area.

**Review Assessment: Checking Correctness Of Derivations And Theory:**

I did not assess the derivations or theory.

**Review Assessment: Checking Correctness Of Experiments:**

I assessed the sensibility of the experiments.

**Review Assessment: Thoroughness In Paper Reading:**

I read the paper at least twice and used my best judgement in assessing the paper.

---

> ### Author Response · Authors · 2019-11-09
> **Response to Reviewer #3**
>
> We would like to thank the reviewer for your constructive feedback.
>
> The reason the current leaderboard looks different from the one we quoted in our paper is that the organizers of the KITTI dataset updated their evaluation protocol on Oct 8th, 2019, which was after the ICLR submission deadline. The results following the evaluation protocol we have used can be found in the following link, and our approach indeed achieves the state-of-the-art.
> http://www.cvlibs.net/datasets/kitti/backups/2019_10_08_10_46_41_object.html
>
> For transfer learning, we have added the comparison with PredNet in Table 5. As you can see, the performance of PredNet on object detection is much worse than our results. We think this is because: (a) PredNet doesn’t have the information preservation property and pixel-wise alignment to give a better localization of bounding boxes. (b) Unlike attention module in our CrevNet, there is no specific layer or module in PredNet that focuses on motions. Motion information can tell the shape of moving objects and distinguish the moving objects from the background.
> However, we are currently unable to evaluate the effectiveness of features learnt from resolution-preserving methods like ContextVP. This is because the resolution of input sequence in the setting of object detection (416*128) is much larger than that in the setting of video prediction (160*128) and resolution-preserving model cannot be fitted into GPUs due to the large memory demand. Overall, we think that the state-of-the-art results on KITTI detection strongly suggest good quality of the learned features.

---

### Official Review · AnonReviewer1 · 2019-10-27
**Official Blind Review #1**

**Rating:** 3

**Review:**

This paper proposed CrevNet, a conditionally reversible network, that performs video prediction efficiently and effectively.
The proposed CrevNet used the bijective two-way autoencoder to make sure the frames are inherently invertible from the latent space.

While the idea of the paper is good, I am not convinced of some points raised in the paper. And I hope to get response from the authors.
(1) The latent space of the proposed method is not very small, it should be at least the same dimension as the input given invertibility, so it is w*h*c. What are the dimensions of resolution-preserving methods?
(2) According to previous studies, although i-RevNet preserves all information in the latent features, it is not make robust mappings: a small perturbation on the features will ruin the inverted image. Is it still the case in CrevNet? Can we some how visualize what is happening/changing in the latent space?

**Experience Assessment:**

I have read many papers in this area.

**Review Assessment: Checking Correctness Of Derivations And Theory:**

I carefully checked the derivations and theory.

**Review Assessment: Checking Correctness Of Experiments:**

I carefully checked the experiments.

**Review Assessment: Thoroughness In Paper Reading:**

I read the paper thoroughly.

---

> ### Author Response · Authors · 2019-11-09
> **Response to Reviewer #1**
>
> We thank reviewer for your feedback.
>
> --Feature size: Yes, your understanding of our volume-preserving architecture is correct. The volume of feature map will always equal to w*h*c, where c=3. For example, if the input size is 128*128*3= 49152, the feature size may be 16*16*192=49152. On the contrary, the dimension of resolution-preserving methods is w*h*k, where w and h will remain the same as those of the input and k usually ranges from 64 to 256. So the size of intermediate features in these kind of methods is at least 20 times larger than that in our CrevNet.
>
> --Robustness: We are not sure if we understand this point correctly, and references to examples of this issue would be helpful. If R1 is referring to the point discussed in section 4.1 of the original i-RevNet paper, this may be a misunderstanding. In section 4.1 the authors argue that the normal implicit inversion can be sensitive to the small variation of input (instead of feature) while the explicit inversion of i-RevNet is robust. Alternatively, R1 may be referring to robustness of the affine coupling layers. The inversion of affine coupling layers might be unstable because it involves division, but we only use additive coupling layers, which are much more stable, in our two-way autoencoder. In our experiments, we never observed any such ruined image. So we think we can claim that our method is robust. As an external support of our claim, there are other published papers like RevGAN [1] and GLOW [2] that also used the reversible architecture for generative tasks.
>
> Should the reviewer feel doubtful about robustness of our proposed model, please see the attached source code for reproducing our experiments.
> https://drive.google.com/file/d/1koVpH2RhkOl4_Xm_q8Iy1FuX3zQxC9gd/view?usp=sharing
>
> --Visualization: Since our method is volume-preserving, we can easily visualize the latent features by reshaping them into the dimension of input.  The visualization of the latent features is provided in the following link. Note that all features are rescaled to [0,1].
> https://drive.google.com/file/d/1JPoDPqnqYeP8PHJ-DJfJXC0AI664C7o5/view?usp=sharing
> Especially, the difference maps demonstrate the changes of latent features. As you can see, the changes of latent features are usually big and the predictions of our CrevNet look fine, which indicates that our method is not sensitive to small pertubation.
>
> Finally, we would be curious to hear R1’s thoughts on our experiments, which show our method achieved the new state-of-the-art on a number of video prediction tasks and demonstrate great transferability of our self-supervised learning method.
>
> [1] van der Ouderaa, T.F. and Worrall, D.E., 2019. Reversible GANs for Memory-efficient Image-to-Image Translation. In Proceedings of the IEEE Conference on Computer Vision and Pattern Recognition (pp. 4720-4728).
> [2] Kingma, D.P. and Dhariwal, P., 2018. Glow: Generative flow with invertible 1x1 convolutions. In Advances in Neural Information Processing Systems (pp. 10215-10224).

---

### Decision · Program_Chairs · 2019-12-19

**Decision:**

Accept (Poster)

**Comment:**

This paper introduces a new approach that consists of the invertible autoencoder and a reversible predictive module (RPM) for video future-frame prediction.

Reviewers agree that the paper is well-written and the contributions are clear. It achieves new state-of-the-art results on a diverse set of video prediction datasets and with techniques that enable more efficient computation and memory footprint. Also, the video representation learned in a self-supervised way by the approach can have good generalization ability on downstream tasks such as object detection. The concerns of the paper were relatively minor, and were successfully addressed in the rebuttal.

AC feels that this work makes a solid contribution with well-designed model and strong empirical performance, which will attain wide interests in the area of video future-frame prediction and self-supervised video representation learning.

Hence, I recommend accepting this paper.